# Method for Measuring the Surface Velocity Field of a River Using Images Acquired by a Moving Drone

**Kwonkyu Yu** [1,*] and **Junhyeong Lee** [2]

1 Department of Civil Engineering, Dong-eui University, Busan 47340, Republic of Korea
2 Department of Civil and Environmental Engineering, Myong-ji University, Yongin 17058, Republic of Korea
* Correspondence: pururumi@gmail.com; Tel.: +82-51-890-1631

**Abstract:** Hovering drones use ground control points to measure the surface flow velocity of rivers. This study aims to use only GPS data and images captured by a drone to extract the flowrate at a designated absolute position. Using GPS data, the moving directions of the drone and of the image were calculated, and each image point was converted into a physical UTM (Universal Transverse Mercator) coordinate system. After determining the range of observation by selecting the start and end frames, all images of the measurement cross section were divided into reference frames (measurement subsections), and the flowrate was calculated with spatiotemporal volume obtained by gathering images for 1 sec (30 frames) for all measurement subsections. The results were comparable with those obtained using the existing hovering drone image analysis method.

**Keywords:** drone; global positioning system; ground control point; spatiotemporal volume; surface image velocimetry; velocity measurement





## 1. Introduction

### 1.1. Motive and Purpose of the Study

The flow velocity of rivers is basic data essential for water resource planning and river disaster prevention. Hence, flow rate measurements are always required, whether during flood or drought seasons. However, measuring the flow rate during floods is a very difficult task, for various reasons. Currently, flood flow measurement in Korea mainly relies on methods using current meters and floats. Measurement using a current meter requires expensive equipment and has the drawback of requiring excessive manpower. Measurements using floats are prone to errors caused by user bias and environmental factors. Because of this, there has recently been an increase in the active study regarding calculating the flow rate with surface image velocimetry obtained from recorded images of the river surface [1].

A surface image velocimeter (SIV) uses a camera for recording images, and therefore, it is inexpensive, requires less manpower, and can instantly measure discharge over a large area [2]. This measurement method is referred to in the literature as large-scale particle image velocimetry (LSPIV); however, in this paper, we will call it 'surface image velocimetry,' referencing its ability to measure velocity using the surface image of a river. Because of these characteristics, SIV has the advantage of being economical and safe. A range of cameras, including camcorders, thermal imaging cameras (far-infrared cameras), smartphone cameras, and CCTVs can be used as the image capturing equipment for SIV [2].

Drones are easier to operate than aircraft and have high resolution over time and space, making them suitable for measuring the physical quantities of rivers. Another notable advantage of drones is that they can easily measure the flow velocity in places that are difficult to access via a camera, such as around river structures or in overflowing areas of weirs. In addition, drones can acquire images with little distortion of perspective through built-in software and a piece of hardware called a gimbal. In addition, by using thermal

imaging cameras, which can be mounted on the drone, images can even be captured at night. Therefore, the drone has a high potential for measuring river flow.

However, these methods of measuring flow velocity using drone images have one unavoidable disadvantage, in that they require ground control points (GCPs) to help them locate the absolute positions of the ground points. This makes it hard to use drones to measure the flow velocity of large rivers, in which it becomes difficult to use GCPs. To some extent, this problem can be solved by increasing the drone altitude, but in this case, the resolution of the image decreases, thereby reducing the accuracy of the flow velocity measurement.

This study presents a method to determine the physical locations (global coordinates) of measurement points in an image, using only the location information of the drone stored in the video. This method was tested in a large river to estimate the flow velocity of the points through image analysis, where the ground control points were not visible. We call this approach the 'moving drone method.'

### 1.2. Flow Velocity Measurement with the Existing Drone Method

Attempts to measure flow velocity began before the existence of drones and used aerial imagery. Kinoshita [3] of Japan calculated the surface flow velocity in 1967 using the Cameron effect (stereoscopic parallax) of two aerial photos of the river surface during flooding. Since then, various studies have been conducted. Studies directly related to this study, using helicopters or drones, include those by Takehara et al. [4] and Kunida et al. [5]. In their studies, it is stated that the shaking of the image was corrected using GCPs on the ground, but it is not specified which method was used. Studies using drones for water resource research in earnest started with Notoya et al. [6]. This study described that the SIFT, RIPOC, and RANSAC algorithms were combined for the first time to track GCPs. Outside Japan, Tauro et al. [7] and Bolognesi et al. [8] attempted measuring the surface flow velocity using a drone, but these studies do not include details about image processing.

Research on drone image velocimetry has been conducted in the other countries. For example, De Schoutheete et al. [9] took a series of images using a drone flying over a natural river and estimated the velocity fields with PIV software. In their study, they stabilized the images using four fixed points on the images. Koutalakis et al. [10] captured a video over a specific area of the Aggitis River in Greece and analyzed it with three different methods: PIVLab, PTVLab, and KU-STIV. When they analyzed the video, they used markers to specify the region of interest. Recently, Fairley et al. [11] applied a drone-based LSPIV to a tidal stream for energy resources assessment. In this study, they used some georeference points to rectify and stabilize the images. Each of these studies largely focused on efficiently locating GCPs and using them to correct images.

In Korea, studies by Yu and Hwang [12] and studies by Liu [13] and Lee et al. [14] can be cited. Liu [13] attempted to find and apply a method for obtaining the physical coordinates of an image without the presence of GCPs, which is the core subject of this study. However, the method demands drone flight log data, which requires another stream of data from the logger, aside from just the images. This also limits the analysis of images recorded in the past, where the availability of flight log data is uncertain. Lee et al. [14] also suggested a method using only the scale of the drone camera lens, but it proved difficult to accurately determine the physical coordinates of a specific location in the image using that approach.

## 2. Materials and Methods

### 2.1. Drone and Camera Used in the Study

The drone used in this study was a Matrice 300 RTK from DJI, with a Zenmuse H20T camera installed. Table 1 shows the appearance and specifications of the drone and the mounted camera.

**Table 1.** Drone and camera used in the study.

| Matrice 300 RTK | |
|---|---|
| Dimension | $810 \times 670 \times 430$ mm |
| Weight | 3.6 kg |
| Maximum hovering time | 55 min |
| Hovering accuracy | Vertical: $\pm 0.5$ m<br>Horizontal: $\pm 1.5$ m |
| **Zenmuse H20T** | |
| Sensor | 1/2.3″ CMOS: $6.16 \times 4.55$ mm$^2$ |
| Maximum image size | $1920 \times 1080$ pixel |
| Lens | FOV 82.9°, 24 mm |
| Gimbal | 3 axes (pitch, roll, yaw) |

### 2.2. Physical Locations in Drone Images

Finding the physical location using drone images requires some additional information. Existing research on surface velocity measurement using drones relies on ground landmarks; the existence of ground control points was essential for correcting the motion image, or for converting image displacement into physical displacement information. However, in large rivers, it is not possible to secure an appropriate number of surface landmarks. Therefore, in this study, a method to determine the physical location in drone images without the use of ground landmarks was devised.

Most drones record flight information in the form of log files in the drone's control program, or in the drone itself. Although this information informs the center of the image, it is not known to what the other parts of the image correspond. In this research, a method was devised to determine the absolute position of each point on the image using the drone's GPS information.

First, the relationship between the image coordinates $(c,r)$ and the local physical coordinates $(x,y)$ for the center of the image is given as follows (refer to Figure 1a).

$$\begin{pmatrix} x \\ y \end{pmatrix} = S_p \begin{pmatrix} 1 & 0 \\ 0 & -1 \end{pmatrix} \begin{pmatrix} c \\ r \end{pmatrix} + S_p \begin{pmatrix} -c_c \\ r_c \end{pmatrix} \tag{1}$$

$(x, y)$ is the local coordinate (unit—m, m), $S_p$ is the transformation coefficient (m/pixel) that converts the image coordinate to the local physical coordinate—and is the reciprocal of the scale—and $(c_c, r_c)$ is the center coordinate (unit—pixel, pixel) of the image.

The transformation coefficient for the camera used in this study, $S_p$, can be expressed as follows [14].

$$S_p = (0.654264\, h + 0.180984) \times 10^{-3}, \tag{2}$$

where h is the vertical height (m) from the water surface to the camera.

In Equation (1), the inverse transformation from image coordinates to local coordinates is as follows.

$$\begin{pmatrix} c \\ r \end{pmatrix} = \frac{1}{S_p} \begin{pmatrix} 1 & 0 \\ 0 & -1 \end{pmatrix} \begin{pmatrix} x \\ y \end{pmatrix} + \begin{pmatrix} c_c \\ r_c \end{pmatrix} \tag{3}$$

where $(c, \text{r})$ is the image coordinate system in units of (pixel, pixel).

To investigate the relationship between local coordinates $(x, y)$ and global coordinates $(E, N)$, it is necessary to know the direction of the drone's image; i.e., the direction which the upper part of the image is facing, which we will call 'video direction.' Video direction must be extracted from the drone's moving direction or azimuth $(\varphi)$, which is the clockwise angle with respect to true north, with values between $-180°$ and $180°$ or $0°$ and $360°$. This study uses values between 0 and $360°$ in the clockwise direction from true north.

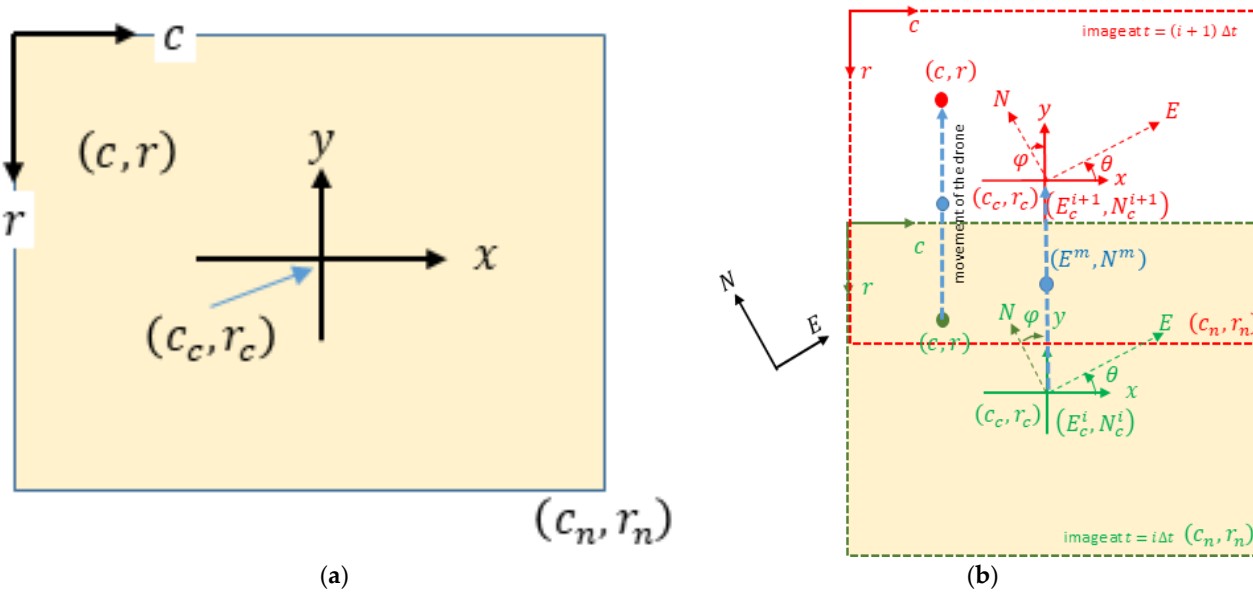

**Figure 1.** Coordinates systems used: (**a**) image coordinates and local physical coordinates; (**b**) local physical coordinates and global coordinates.

For the $i$-th image, a counterclockwise rotation of the local coordinates $(x, y)$ by $\varphi$ and translation by $(E_c^i, N_c^i)$, the global coordinate of the center of the $i$-th image, results in a change from local coordinates to global coordinates, as shown in Figure 1b. The conversion equation is as follows.

$$\begin{pmatrix} E \\ N \end{pmatrix} = \begin{pmatrix} \cos\varphi & -\sin\varphi \\ \sin\varphi & \cos\varphi \end{pmatrix} \begin{pmatrix} x \\ y \end{pmatrix} + \begin{pmatrix} E_c^i \\ N_c^i \end{pmatrix} \tag{4}$$

Conversely, the inverse transformation that converts the global coordinates in Equation (4) into local coordinates is as follows.

$$\begin{pmatrix} x \\ y \end{pmatrix} = \begin{pmatrix} \cos\varphi & \sin\varphi \\ -\sin\varphi & \cos\varphi \end{pmatrix} \begin{pmatrix} E - E_c^i \\ N - N_c^i \end{pmatrix} \tag{5}$$

Therefore, Equations (3) and (5) can be used to represent the transformation from global coordinates to image coordinates, and Equations (1) and (4), to represent the transformation from image coordinates to global coordinates.

The image direction $\varphi$ must be known in the above transformation relationship. The video direction can be obtained from the GPS information recorded by the moving image of the drone. An important assumption we make is that the drone moves at a constant speed, and only in the image direction (r axis) or downward direction (r axis). In addition, the image direction is expressed as the azimuth, $\varphi$. However, it is difficult to obtain the image direction directly, and therefore, it must be obtained using the moving direction of the drone, $\theta$ (counterclockwise angle from east). $\theta$ can be obtained using the physical coordinates of the center of the 0-th image (start frame), $(E_c^0, N_c^0)$, and the physical coordinates of the center of the $n$-th image (end frame), $(E_c^n, N_c^n)$, recorded in the image.

$$\theta = \begin{cases} \tan^{-1}\left( \dfrac{N_c^n - N_c^0}{E_c^n - E_c^0} \right), & (E_c^n > E_c^0) \\ \tan^{-1}\left( \dfrac{N_c^n - N_c^0}{E_c^n - E_c^0} \right) + \pi, & (E_c^n < E_c^0) \end{cases} \tag{6}$$

$\left(E_c^0, N_c^0\right)$ denotes the physical coordinates of the center of the 0-th image (start frame), and $\left(E_c^n, N_c^n\right)$ denotes the physical coordinates of the center and the $n$-th image (end frame), respectively.

This relationship is detailed in Figure 2. The image direction $\varphi$ and the moving direction $\theta$ of the drone have a relationship of $\varphi = \frac{\pi}{2} - \theta$.

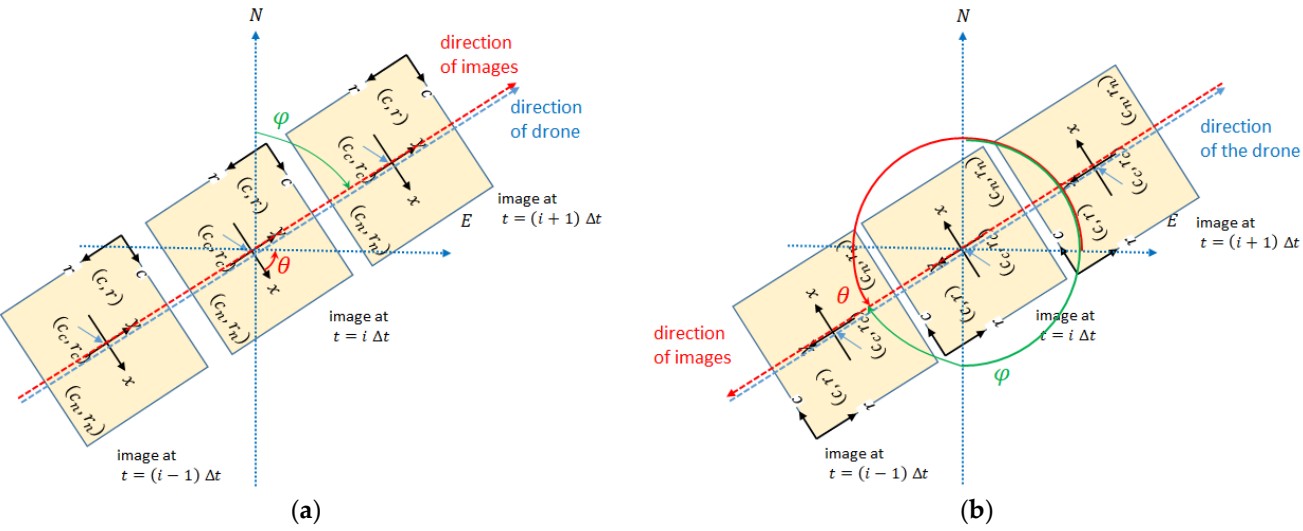

**Figure 2.** Image direction and the drone's moving direction: (**a**) drone moving in the same direction; (**b**) drone moving in the opposite direction.

### 2.3. Spatiotemporal Images for Each Measurement Section

The process of creating spatiotemporal images for each measurement subsection and subsequently analyzing the flow velocity is explained by Figure 3. As with Figure 3①, the measurement subsections are determined. To do this, the start and end frames and their corresponding physical coordinates are calculated by the method explained earlier. Then, images of the measurement subsections located equal distances from one another in the global coordinates are analyzed. The frame count that includes the subsection is dependent on the scale of the image and the drone's velocity, but to accurately calculate surface flow velocity using the spatiotemporal image analysis method, at least 1 sec (30 frames) of images are required. (Tests done during this study proved 30 frames sufficient in all conditions.). Therefore, the frame where the $i$-th subsection is exactly (or closest) to the center of the image is taken as the $i$-th reference frame. The reference frame, 15 consecutive frames before the reference frame, and 14 images after the reference frame, for a total of 30 images, are collected and analyzed (Figure 3③). The 30 images extracted for each subsection are moved such that the horizontal center line of the reference frame and the image coincide based on the moving speed of the drone (Figure 3④). By forming a grid at regular intervals on this horizontal center line and analyzing the spatiotemporal volume, the flow field can be obtained (Figure 3⑤). For details of the image analysis method using spatiotemporal volume, refer to Yu and Liu [12].

The drone surface image velocity meter program was written in Python, with module PyQt6 being used for GUI, OpenCV for image processing, Pandas for data input/output, and pyproj for WGS and UTM coordinate transformation.

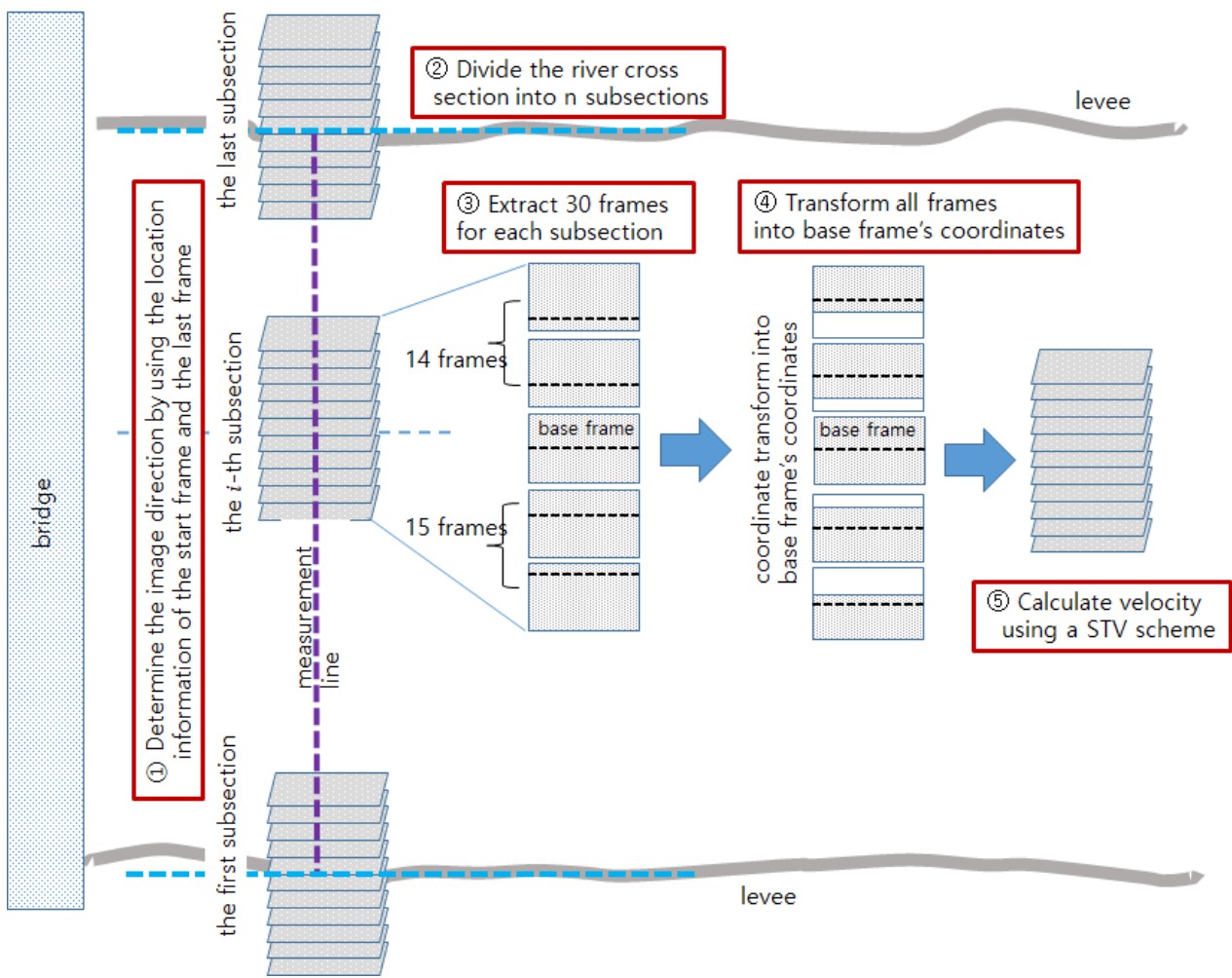

**Figure 3.** Schematic of the navigating drone method.

## 3. Results

### 3.1. Andong River Experiment Center's Steep Waterway

The program developed in this study was tested on the steep slope of the River Experiment Center in Andong, Korea. The test used a video filmed on 15 April 2021, long before the development of the program. Figure 4 shows the movement directions of the test site and the drone.

Figure 5 shows a few frames of this video, which have a size of 1920 × 1080 pixels, and the drone's recorded GPS information is shown in the upper left area of the video. The recording frequency of the GPS information was 0.1 s, and it provides the latitude, longitude, and altitude, in sequence.

In Figure 5, the position of the drone is shown on the upper left, in the order of latitude, longitude, and altitude. However, since these data are difficult to apply to actual calculations, they must be converted to UTM coordinates. For coordinate conversion between WGS and UTM, Python's pyproj module was used. The conversion condition is "ellipsoid: GRS80, origin: central, origin addition: after 2009.12." For WGS and UTM, "epsg:4326" and "epsg:5186" were selected, respectively.

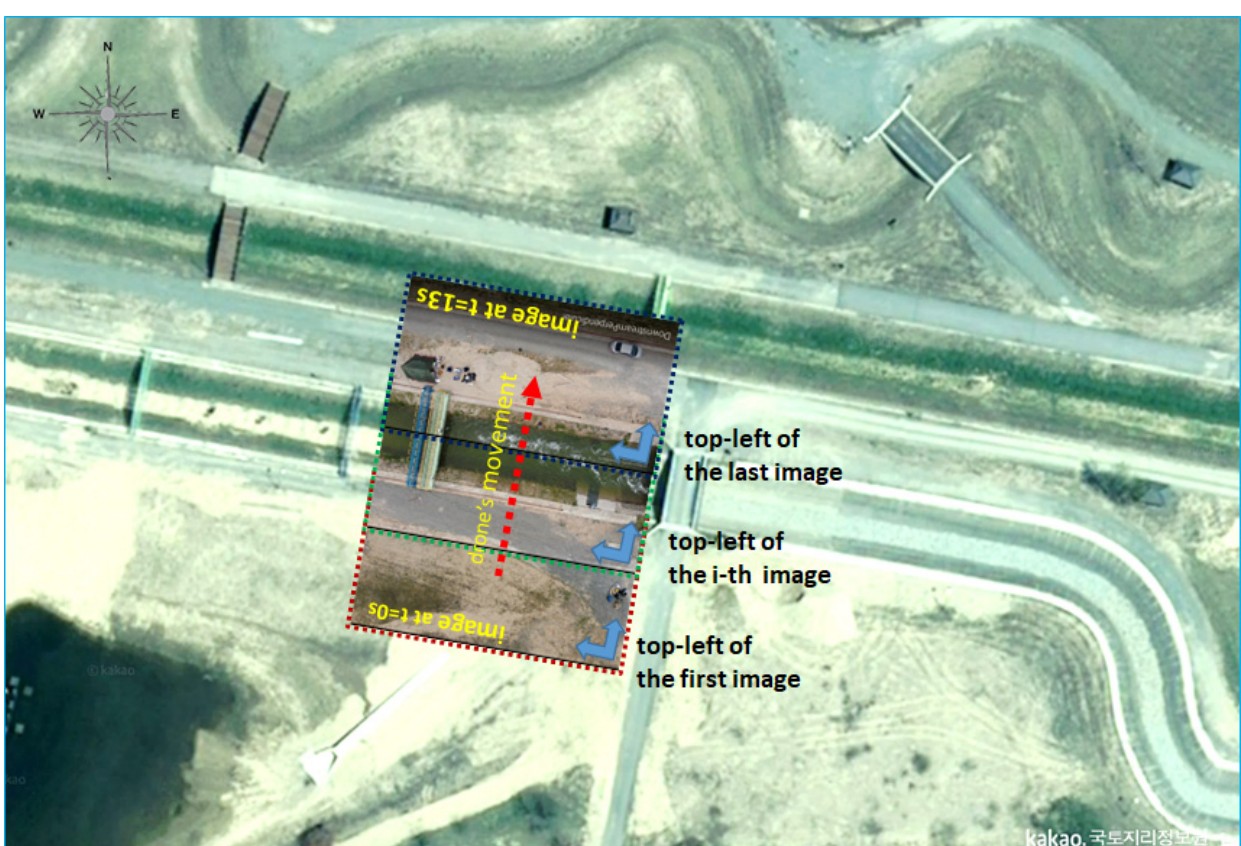

**Figure 4.** Test site (River Experiment Center) and the drone's flight path.

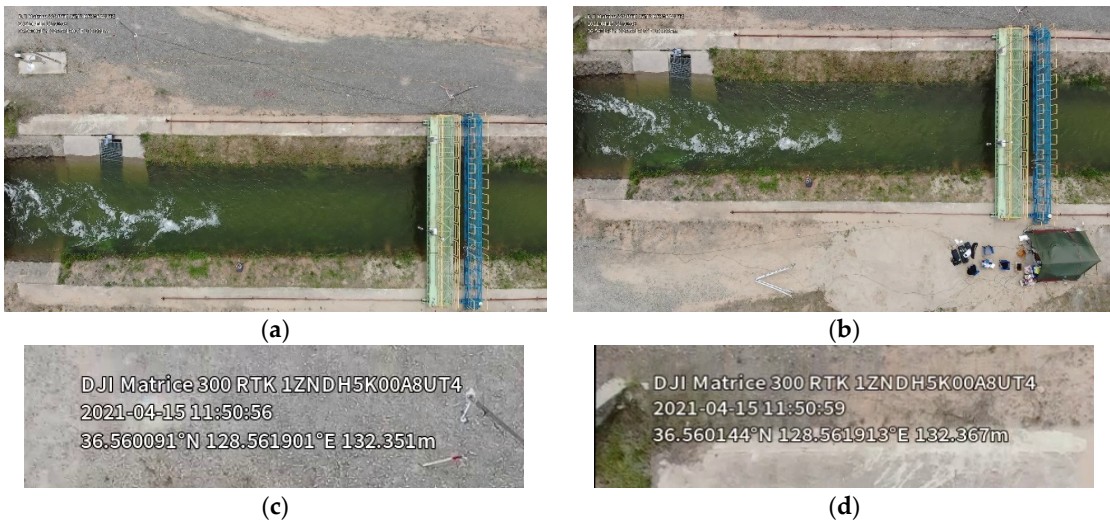

**Figure 5.** Examples of frame images captured with the drone camera: (**a**) frame 180; (**b**) frame 269; (**c**) top-left area of frame 180; (**d**) top-left area of frame 269.

The GPS coordinates included in the video were acquired using Python's EasyOCR module. Like most other DJI drones, the Matrice 300 RTK used in this study measures and records the GPS information of the drone once every 0.1 s [10]. It can be seen that the drone's flight path was a straight line along the given route, and it did not show any irregularities. The drone proceeded in the downward direction of the image at a speed of 1.96 m/s.

However, since the first and last segments were difficult to use because of a course alteration, only the frames in between were used. To compare the results with those from the existing methods using the image correction technique based on fixed points, approximately 6 s (185 frames) of the video, from frame 140 to frame 325 in which a clear fixed point can be observed, were analyzed. As a result of examining the video in Figure 5 using the method presented in Figure 2, the moving direction of the drone was $\theta = 79.63°$, and the video direction was $\varphi = 259.63°$. A total of 11 grid points at 2.0 m intervals in the horizontal direction, and 14 grids at 1.0 m intervals in the vertical direction were used.

The results from the above method are shown in Figure 6. Although it is difficult to confirm visually, the flow velocity of the ground is calculated to be approximately 0, and the flow velocity distribution is appropriate along the cross section of the waterway. In addition, Figure 6 shows the results of processing the same video using the hovering drone method of converting all frames to reference frames using ground vertices and analyzing the images. However, in the hovering drone method, 90 frames (3 s video) from frame 180 to frame 269 were used because the images shared four points that can be used as ground control points.

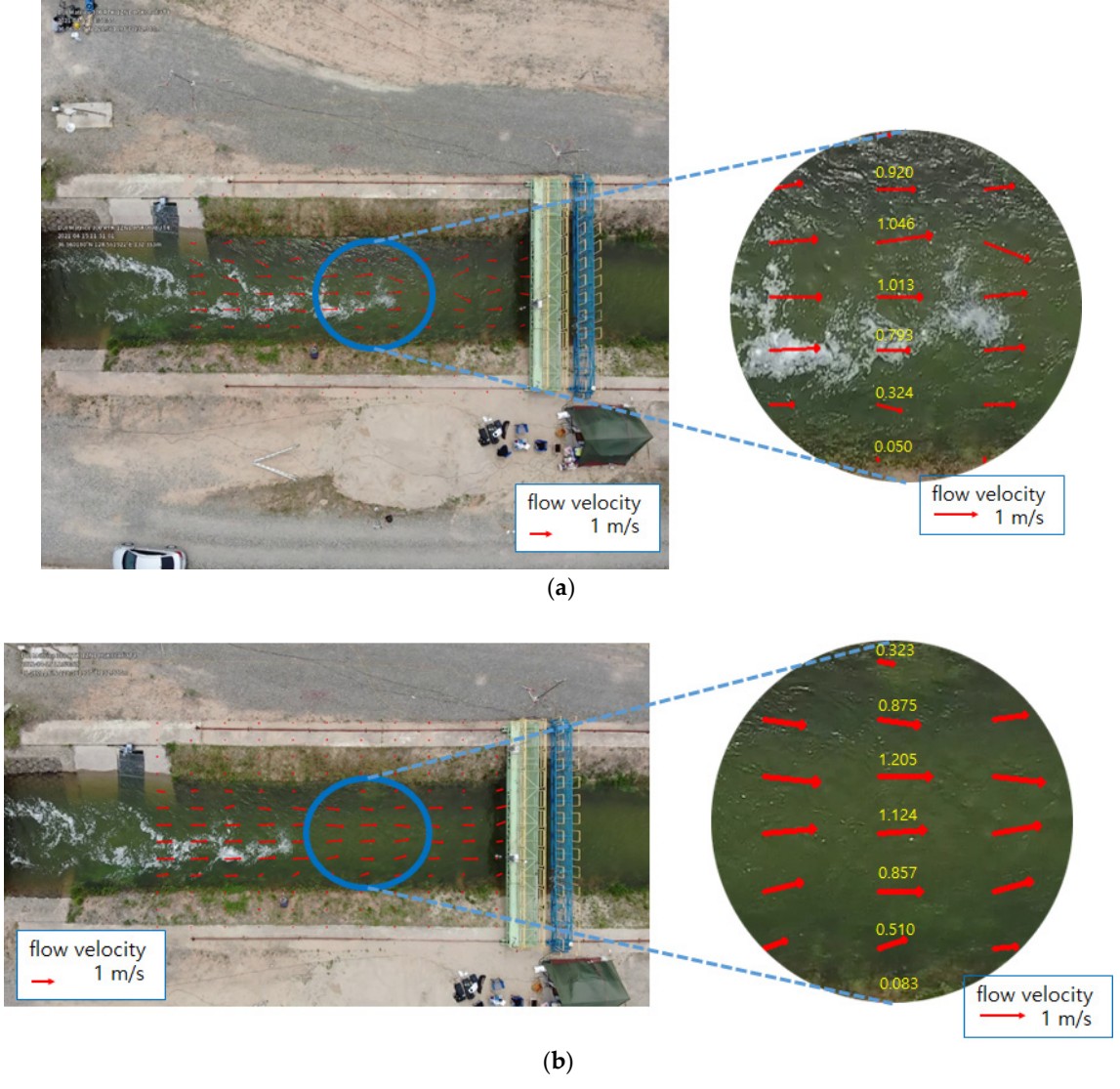

**Figure 6.** Velocity fields measured using the drone images. (**a**) Moving drone method; (**b**) Hovering drone method. (the numbers representing flow velocity were added later for reference).

For the numerical comparison, Figure 7 shows the flow velocity in the center of the image and the velocity measured with a radio anemometer near the second section on the downstream side of the image, at the downstream measurement stand. Because of the difference between the three methods, it was not possible to measure the flow velocity at the exact location; a difference of approximately 0.34 m is found in the vertical direction. The results are corrected to similar locations and are presented below.

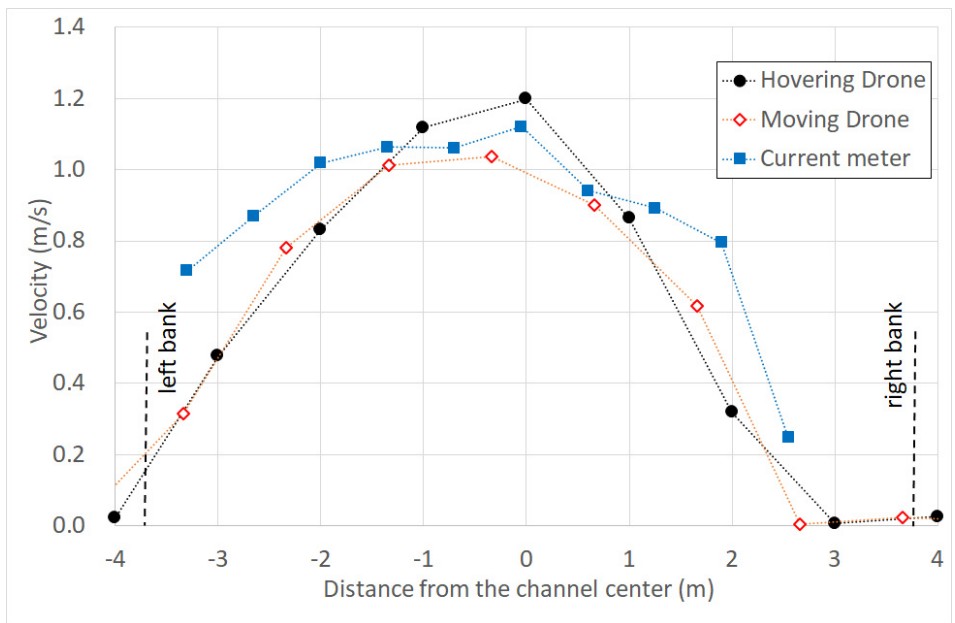

**Figure 7.** Velocity distribution (Andong Experimental Station).

From this result, the flow velocity shows a difference of up to 0.3 m/s at one point, but the flow velocity at other measurement points does not show a large error, and it shows a similar value. The microwave current meter shows significantly less change across the cross section than does the image flow rate. This seems to be due to the characteristics of the microwave current meter. This uncertainty occurs because the detection area is large, preventing the measurement position from being accurately indicated. Therefore, it seems better to recognize the values presented in Figure 7 only as reference data.

### 3.2. Mujudaegyo Bridge Upstream

The final application goal of the method developed in this study is to measure a site where the ground control points cannot be installed because the width or size is too wide or too large. To that end, the developed method was applied to the upstream region of the Mujudaegyo Bridge, which is up to 140 m wide. The on-site situation and video recording using drones are shown in Figure 8. As Figure 8 shows, the river was too wide to allow for the capture of an adequate number of ground control points for both sides of the river into one frame.

The video was filmed from an altitude of 50 m over the 30 m upstream area of the Mujudaegyo Bridge, from the left bank to the right bank, and then from the right bank to the left bank, at a flight speed of 2.0 m/s. However, since the first and last segments are difficult to use due to the changed course, 2048 frames in the middle (frame 1252 matching the left bank, frame 3300 reaching the right bank) were used for the northward flight image, which was acquired for 68 s. Similarly, 2048 frames (from frame 3522 on the right bank to frame 5543 on the left bank) were used for the south direction flight image. As a result of examining the video in Figure 5 using the method presented in Figure 2, the moving direction of the drone was $\theta = 105.1°$, and the video direction was $\varphi = -15.1°$. In this case,

11 grid points at 2.0 m intervals in the horizontal direction and 29 grid points at 5.0 m intervals in the vertical direction were used as the grid.

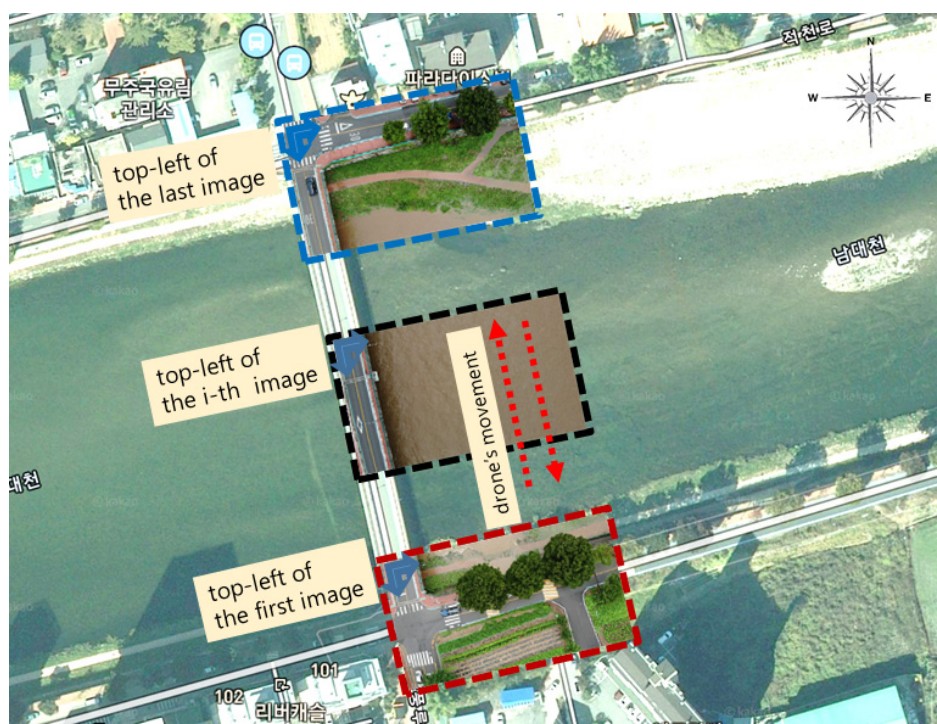

**Figure 8.** Test site (Mujudaegyo Bridge) and the drone's flight path.

The results analyzed using the above method are shown in Figure 9. Although it is a little difficult to visually confirm this result, the velocity on the ground is approximately 0, and the flow velocity distribution is appropriate along the channel cross section.

Although there is a slight difference, other than the low flow rate, most of the results show an error within 5%, which is quite satisfactory. The flow velocity data measured with a microwave current meter—at approximately the same time when this video was taken—on the railing of the Mujudaegyo Bridge, which is located 22 m, 50 m, and 75 m away from the left bank, are compared with the velocity data calculated from the drone image, as shown in Table 2.

When we use the velocity measured with the microwave current meter, we must keep in mind that the microwave spreads at an angle of 15 degrees. Considering that the distance between the measurement point and the current meter is about 15 m in this experiment, the measurement area at the measurement points would be about 7.5 m × 7.5 m. It is considerably wider compared with the channel, and it is difficult to improve the accuracy of the results, even if the grid size is increased, since the grid size only makes the searching area larger. In order to increase the accuracy of the velocity measurement using UAVs, we have to lower the drone's altitude, that is, decrease the distance from the water surface.

**Table 2.** Comparison of velocities measured.

| Microwave Current Meter | | Navigating Drone (South → North) | | Navigating Drone (North → South) | |
|---|---|---|---|---|---|
| Dist. (m) | Velocity (m/s) | Dist. (m) | Velocity (m/s) | Dist. (m) | Velocity (m/s) |
| 22 | 3.07 | 25 | 2.76 | 25 | 2.99 |
| 50 | 3.66 | 50 | 3.22 | 50 | 3.18 |
| 75 | 3.20 | 75 | 3.20 | 75 | 3.20 |

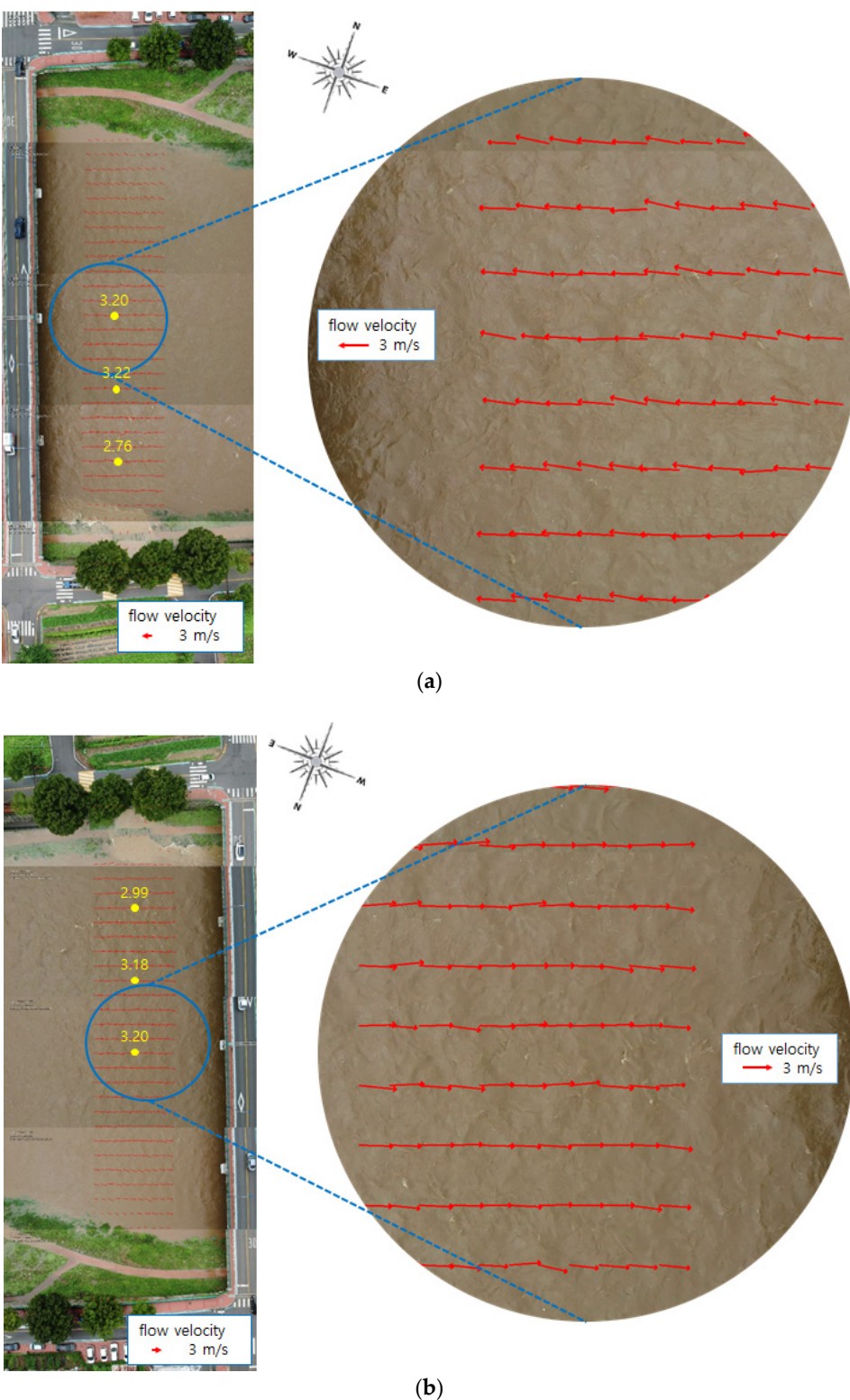

**Figure 9.** Velocity fields (Mujudaegyo Bridge) measured with the navigating drone method: (**a**) flight from south to north; (**b**) flight from north to south. (The numbers representing flow velocity were added later for reference.)

Furthermore, the outcome of directly displaying this result on Google Maps using QGIS is shown in Figure 10. Comparing Figure 10 with Figure 9a, it can be seen that the calculated flow velocity distribution is properly displayed on Google Maps.

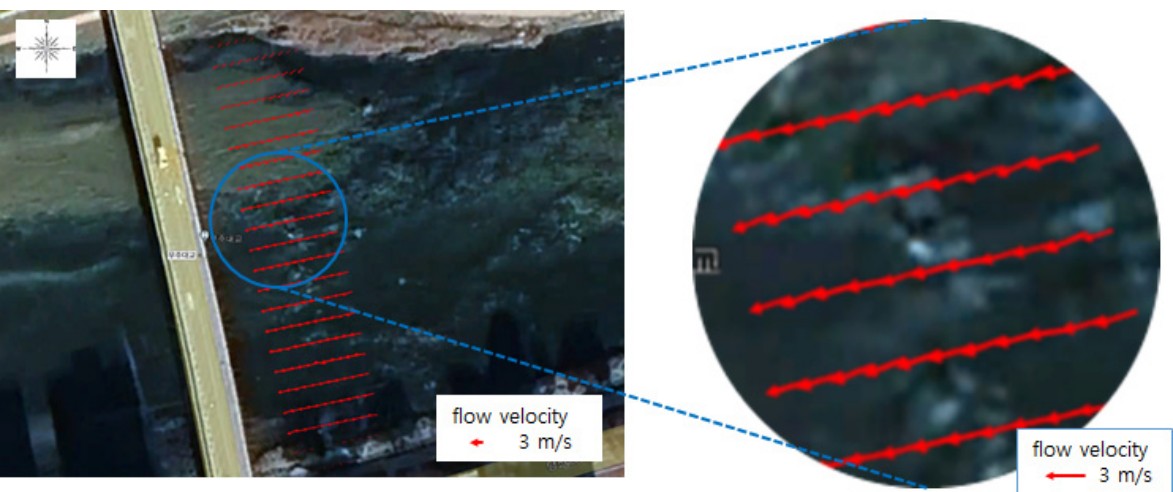

**Figure 10.** Velocity fields (Mujudaegyo Bridge) plotted in Google Maps.

## 4. Conclusions

We developed a method to calculate the flow velocity of an absolute position in a drone video, using only the location information (GPS data) recorded in the video, with no ground coordinate points.

1. A method of determining the moving direction of the drone and the direction of the image using GPS information in the drone video is presented. To this end, the image direction is first determined, and then each image point is converted into a physical global coordinate system (UTM coordinate system).
2. The start and end frames are designated, then divided into reference frames (measurement subsections) at regular intervals. Images are collected for 1 s (30 frames), and the flow rate is calculated using spatiotemporal volume analysis.
3. The analyzed image is displayed on the integrated total image. In addition, the analysis data can be displayed directly on Google Maps.
4. The comparison of our method to the existing method of using video analysis on the footage from a stationary flying drone, was very favorable.

The results of this study can be widely used to measure the surface flow field in a wide body of water or river, where a reference point cannot be installed. This method works using only drone images, and no additional surveying work is required.

**Author Contributions:** Conceptualization, K.Y. and J.L.; methodology, K.Y. and J.L.; formal analysis, K.Y. and J.L.; investigation, J.L.; data curation, J.L.; writing—original draft preparation, K.Y.; writing—review and editing, K.Y.; visualization, K.Y. and J.L. All authors have read and agreed to the published version of the manuscript.

**Funding:** This research was funded by Korean Ministry of Environment (MOE) (2020003050002).

**Institutional Review Board Statement:** Not applicable.

**Informed Consent Statement:** Not applicable.

**Data Availability Statement:** This study did not include any publicly available datasets.

**Acknowledgments:** This work was supported by the Korea Environmental Industry & Technology Institute (KEITI) through the Aquatic Ecosystem Conservation Research Program, funded by the Korean Ministry of Environment (MOE) (2020003050002).

**Conflicts of Interest:** The authors declare no conflict of interest.

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
