# Peer review of "Method for Measuring the Surface Velocity Field of a River Using Images Acquired by a Moving Drone"

_water, doi:10.3390/w15010053_

Round 1
Reviewer 1 Report
The method presented in this work calculates velocity fields in moving rivers. The work's main contribution is the definition of reference points around the river relative to a drone's movement. Then, the reference points are used to measure the velocity fields of the moving rivers. The work is interesting but difficult to follow since there is a lack of information; because two key references in the manuscript are important and written in Korean, I guess not everyone understands Korean. Therefore, the authors should discuss the procedure to measure the velocity field in depth and how equation 2 [9, 11] was obtained in this manuscript.
On the other hand, I suppose that measurements are realized in very calm conditions, that is, no wind that perturbs the drone. However, the author did not say anything about it. ¿How does this condition affect the results reported?
¿Are drone vibrations a problem for these measurements?, Or ¿are they negligible for these applications?
The images obtained were divided into small regions. ¿How many pixels does each area contain?
Figure 7 should be explained more clearly explained. For example:
- ¿Are the measurements with the radio anemometer representative compared to the measures reported in this work?
- Can the measurement area covered by the radio anemometer be compared with the measurement area corresponding to the results presented in this work?
- Increasing the grid size in the area of interest improves the results. What's going on?
Author Response
Response to Reviewer 1 Comments
Thank you for your kind and thorough review.
Point 1. The method presented in this work calculates velocity fields in moving rivers. The work's main contribution is the definition of reference points around the river relative to a drone's movement. Then, the reference points are used to measure the velocity fields of the moving rivers. The work is interesting but difficult to follow since there is a lack of information; because two key references in the manuscript are important and written in Korean, I guess not everyone understands Korean. Therefore, the authors should discuss the procedure to measure the velocity field in depth and how equation 2 [9, 11] was obtained in this manuscript.
Response 1: As the reviewer pointed out, the two key procedures are how to estimate the transformation coefficient by Lee et al. [11], and how to calculate the velocity field using spatio-temporal volume by Yu et al. [9]. To help readers understand, we added the relevant content.
Point 2. On the other hand, I suppose that measurements are realized in very calm conditions, that is, no wind that perturbs the drone. However, the author did not say anything about it. How does this condition affect the results reported?
Response 2: In this study, we use drone’s automatic navigation mode. In this automatic navigation mode, the drone travels along a predetermined path. In this mode, the drone overcomes the wind if it isn’t too strong. The manual of Matrice 300 RTK drone says the maximum endurable wind speed is 15 m/s. We added this to Table 1.
Point 3. Are drone vibrations a problem for these measurements? Or are they negligible for these applications?
Response 3: We think the drone vibration would be negligible in this application, since the spatio-temporal volume and spatio-temporal image analyses are similar to the averaging of image displacement for a given time.
Point 4. The images obtained were divided into small regions. How many pixels does each area contain?
Response 4: The image patch sizes 64×64 pixels. We added this information to the paper.
Point 5. Figure 7 should be explained more clearly explained. For example:
- Are the measurements with the radio anemometer representative compared to the measures reported in this work?
Response 5: As the reviewer knows, microwave current-meter is one of indirect measurement method unlike the traditional propeller current-meter. So using its measurement data might be inappropriate to compare or to check measurement data taken by other instrument. In this case, however, it is nearly impossible to use the traditional propeller current-meter, so we had no choice but to use the microwave current-meter as the only alternative.
- Can the measurement area covered by the radio anemometer be compared with the measurement area corresponding to the results presented in this work?
Response 6: The microwave spreads at an angle of 15 degrees. Considering that the distance between the measurement point and the current-meter is about 5 m in this experiment, the measurement area at the measurement points would be about 2.5 m × 2.5 m. It is considerably wider compared with the channel.
- Increasing the grid size in the area of interest improves the results. What's going on?
Response 7: The authors think it is difficult to increase the accuracy of the results even if the grid size is increased, since the grid size only makes the searching area larger. In order to increase accuracy, we have to lower the drone’s altitude, that is, decrease the distance from the water surface.
Reviewer 2 Report
A method is developed for the calculation of the flow velocity of an absolute position in a drone video using only the recorded location information (GPS data). A favourable comparison of the present method against previous ones is also presented. The paper is well written and I believe it will be interesting to the readers of Water journal. I have no specific comments and I recommend the paper for publication in the Water journal as it is.
Author Response
Point 1. A method is developed for the calculation of the flow velocity of an absolute position in a drone video using only the recorded location information (GPS data). A favourable comparison of the present method against previous ones is also presented. The paper is well written and I believe it will be interesting to the readers of Water journal. I have no specific comments and I recommend the paper for publication in the Water journal as it is.
Response 1: Thank you for your kind review.
Reviewer 3 Report
1.For ''Our results compared favorably with those obtained using the existing hovering drone method. (21)'', it is suggested to provide specific parameter indicators to illustrate the comparison between the method and the existing methods to increase the credibility of the article.
2.For Section 2.1, the logic organization is not strong and the summary is lacking. It is suggested that the author reorganize this section, sort out the writing logic, and summarize the research status of flow velocity measurement using UAV at the end of this section。
3.There are two sections 2.3 in the article, please correct the author and correct the writing attitude.
4.Regarding the 'For details of the image analysis method using spatio-temporal volume, refer to Yu and Liu (181)' proposed in the paper, the author did not give a clear explanation on how to measure the velocity of the wave surface, please add.
5. The method proposed in this paper for the measurement of the velocity field of the river surface lacks error analysis.
6. In Table 2, in comparing microwave galvanometer and UAV navigation speed measurements, the microwave galvanometer uses a distance of 22m, while the UAV navigation distance is 25m. The comparison of the measured results is not convincing.
7.The article lacks innovation: The method proposed by the author just does not require the use of ground coordinates for flow velocity measurement, lacks innovation, and has little impact and promotion on fluid dynamics.
8. The language of the article can be polished.
Author Response
Response to Reviewer 3 Comments
Thank you for your kind and thorough review.
Point 1. For ''Our results compared favorably with those obtained using the existing hovering drone method. (21)'', it is suggested to provide specific parameter indicators to illustrate the comparison between the method and the existing methods to increase the credibility of the article.
Response 1: The most important factor in evaluating the newly developed method is to compare it with the measured flow rate or to the existing method. For this purpose, in this paper, the flow velocity calculated by the newly proposed method was compared with the actual flow velocity and that calculated by the existing method. Though the result seems to have a little bit large errors, we think the results could be acceptable. Since this paper aims to propose new method to estimate flow velocity at a designated absolute position without ground control points and show its applicability, the error analysis would be left to future research.
Point 2. For Section 2.1, the logic organization is not strong and the summary is lacking. It is suggested that the author reorganize this section, sort out the writing logic, and summarize the research status of flow velocity measurement using UAV at the end of this section。
Response 2: The current status of the existing flow velocity measurement method using UAV is summarized at Section 2.1.
Point 3. There are two sections 2.3 in the article, please correct the author and correct the writing attitude.
Response 3: Thank you for your careful point. It’s the wrong section number. We corrected the section number.
Point 4. Regarding the 'For details of the image analysis method using spatio-temporal volume, refer to Yu and Liu (181)' proposed in the paper, the author did not give a clear explanation on how to measure the velocity of the wave surface, please add.
Response 4: We added a paragraph that explain the velocity analysis method using the spatio-temporal volume summarized Yu and Liu’s method[9].
Point 5. The method proposed in this paper for the measurement of the velocity field of the river surface lacks error analysis.
Response 5: Since this paper aims to propose new method to estimate flow velocity at a designated absolute position without ground control points and show its applicability, the error analysis would be left to future research.
Point 6. In Table 2, in comparing microwave galvanometer and UAV navigation speed measurements, the microwave galvanometer uses a distance of 22 m, while the UAV navigation distance is 25 m. The comparison of the measured results is not convincing.
Response 6: Although the positions of microwave current-meter and drones are not exactly agreed to each other in Table 2, the estimated velocities seem to be similar.
Point 7. The article lacks innovation: The method proposed by the author just does not require the use of ground coordinates for flow velocity measurement, lacks innovation, and has little impact and promotion on fluid dynamics.
Response 7: Until now, all flow velocity measuremt using drones have required the ground control points. Therefore, it was impossible to measure the flow velocity of a wide water area where the ground control points were not available using drones. This paper presents a new method to solve this problem and to measure the velocity field even in wide waters without GCPs. This is something that no one has succeeded. The authors think this is a new innovation in this area.
Point 8. The language of the article can be polished.
Response 8: We asked a native speaker to rewrite the whole paper.
Round 2
Reviewer 1 Report
I do not have more comments. The manuscript can be published in the present form.
Author Response
Thank you for kind and thoughtful review.